# A Portable Photocollector for the Field Collection of Insects in Biodiversity Assessment

**DOI:** 10.3390/insects15110896

**Published:** 2024-11-16

**Authors:** Behnam Motamedinia, Sophie Cardinal, Scott Kelso, Carolyn Callaghan, Khorshid Ghahari, John F. Wilmshurst, Jeff Skevington

**Affiliations:** 1Canadian National Collection of Insects, Arachnids and Nematodes, Agriculture and Agri-Food Canada, 960 Carling Avenue, Ottawa, ON K1A 0C6, Canada; sophie.cardinal@agr.gc.ca (S.C.); scott.kelso@agr.gc.ca (S.K.); kghahari@uoguelph.ca (K.G.); 2Canadian Wildlife Federation, 350 Promenade Michael Cowpland Drive, Kanata, ON K2M 2W1, Canada; carolync@cwf-fcf.org (C.C.); jwilmshurst@cwf-fcf.org (J.F.W.); 3Biology Department, Carleton University, 207 Nesbitt Biology Building, 1125 Colonel By Drive, Ottawa, ON K1S 5B6, Canada; jhskevington@gmail.com; 4Centre for Biodiversity Genomics, University of Guelph, 50 Stone Road East, Guelph, ON 10 N1G 2W1, Canada; 5Department of Geography and Planning, University of Saskatchewan, 117 Science Place, Saskatoon, SK S7N 5C8, Canada

**Keywords:** biodiversity, insect sampling, invention, photoeclector, phototropism, separator, sampling efficiency, smart collector

## Abstract

Studies regarding insect biodiversity often require large samples, which are not always easy to obtain since preparing and cleaning the debris from samples takes much time. Insects are usually acquired using sweep netting methods, but these yield a great amount of additional material to sort. With the purpose of lightening this task, we designed a low-cost photocollector device powered by an LED light source for attracting insects. Timed trials were conducted in the grasslands of the Canadian prairies to determine its efficiency in sorting live insects from debris. For this purpose, two group of insects were considered: bees and flies. We noticed that various species of bees and flies moved at different speeds. This would mean our photocollector can serve as an effective tool to accelerate insect collection based on their speed, thereby contributing to the study of insect diversity.

## 1. Introduction

There are an estimated 5.5 million insect species worldwide [1], many of which remain undescribed. Because of their vital ecological roles in pollination, nutrient cycling, and biological control [2], and their presence in almost all terrestrial ecosystems, insects are often used as target taxa in ecological studies (e.g., [3]; see review in [4]). Their diversity and abundance, however, make thorough biodiversity assessments a time-consuming and expensive job.

Ecologists and taxonomists use various methods to collect insects in the field, including pan traps, Malaise traps, sweep nets, and increasingly light traps linked to digital photography and machine learning algorithms (e.g., [5,6]). Despite advances in non-destructive sampling, physically collecting insects remains the standard for biodiversity and taxonomic studies.

Sweep netting is an active method that is common and effective in collecting arthropods associated with vegetation in diverse habitat types [7,8], including native prairies [6], but it is labor intensive [9]. Using sweep nets for non-specific or mass collections typically requires that insect specimens first be separated from unwanted materials and plant debris. Debris sorting becomes a rate-limiting step in the collection workflow. It is generally advisable to separate live insects in the field, rather than using killing agents to secure specimens while they are in vegetation debris because dead insects, particularly small ones, are significantly more challenging to extract from debris compared to their live counterparts [10]. It is particularly important to precisely separate insects from unwanted material that could inhibit sequencing and metabarcoding [11].

Several tools are available for separating live insects from debris in the field. An aspirator, commonly referred to as a “pooter,” is a convenient device to collect live insects from sweep nets. However, specimens often escape, even with electric and portable pooters (see the Elepooter, [12]). The Berlese funnel was designed to extract insects and arthropods from leaf litter and soil that display negative phototropism and is widely used to study arthropods in bird nests and soil [13,14,15]. Positive phototropism is a broad phenomenon and only one aspect lends itself to the development of a light gateway for insects that are naturally diurnal. It has been used to collect insects from vegetation (e.g., [10,16,17]).

Tracing the origins of photocollectors (also referred to as photoeclectors) is difficult, but in describing the new commonly used Malaise trap, Malaise [18] refers to methods that use insects’ attraction to light as being “known of old”. Phototaxis devices were described in the 1960s by Andrzejewska and Kajak [19] and Gromadzka and Trojan [20] but were not used for sifting through sweep net debris, rather for directly collecting insects from vegetation in the field. Augustin Hoffer, a Czech entomologist, is known to have used a type of photocollector in the 1950s and 1960s to sift through sweep net debris (Lubo Masner, personal communication). Furthermore, Dr. Jan Obenberger is reported as using of a type of photocollector for sorting insect samples from debris during the Tenth International Congress of Entomology, Montreal, Canada, 1956 (Lubo Masner, personal communication). This use of photocollectors is also known to have occurred in Cuba by the mid-1980s (Jose Fernandez-Triana, personal communication).

The photocollector is a potentially indispensable tool in entomological studies that remains largely untested and undescribed in the scientific literature. We constructed a photocollector that is portable, lightweight, cost-effective, and equipped with rechargeable light to extract insects from debris in the field. As well, given that insect taxa have different degrees and intensities of movement to light, we assess the photocollector’s effectiveness in removing insect samples from debris using timed trials from 1 to 4 h to determine the maximum time needed to run samples through the photocollector and whether this time varies across taxa.

## 2. Materials and Methods

### 2.1. Design and Construction

Our photocollector was made from a simple black plastic container (35 cm long × 25 cm wide × 14 cm deep) weighing 679 g (we used Ikea’s Uppsnofsasd™ storage box with lid, sourced in Ottawa, ON, Canada) (Figure 1(A1)).

The collection bottle (500 mL threaded plastic, Figure 1(B1)) was secured to the box through a 6 cm diameter hole in the container’s front wall (Figure 1(A3,A4)). Inside the container, we added a ramp (inclined floor) (Figure 1(A2)) made of 6 mm thick hardboard and cut to 20 cm wide × 30 cm long to provide insects access to the entrance of the collection bottle. The free end of the ramp terminated just below the entrance hole to the collection bottle. A bead of hot glue from a hot glue gun was applied along all 4 edges of the ramp to fill in any gaps between the ramp and the sides of the container.

To secure the collection bottle to the container, the bottle cap was removed, and the top of the cap cut off, leaving a threaded ring (Figure 1(B2)). We then cut the soft rubber top of a calf feeder bottle (Figure 1(B3)), stretched the nipple securely over the outside of the threaded ring (Figure 1(E2)), and glued this part to the black container from inside of the 6 cm hole (Figure 1E).

Inside the black container, to guide insects directly into the collection container, we modified the top 12 cm of a 1.5 L water bottle, cut from mouth to shoulder (Figure 1(B4)) to fit into the calf feeder nipple. The water bottle top was painted white and the shoulder was trimmed obliquely to fit in the box with the mouth of the bottle pushed through the small opening of the calf feeder nipple (Figure 1E). The bottom of the bottle shoulder was glued to the ramp and the shoulder walls were then taped with 60 mm Tuck tape to the flexible guide walls made of lightweight packing foam (Figure 1(F1)). This was in turn taped to the side walls of the collection box and the ramp (Figure 1F). The guide walls were tall enough to reach all the way to the lid of the collection box, forming a complete seal to ensure that insects could not easily get into the space behind the guide walls and beneath the ramp.

A larger hole was drilled off center in the box’s lid, approximately 10 cm from one end of the long axis of the lid, centered across the short axis. It is through this hole that the sweep net samples, consisting of insects and debris, were put into the photocollector. A tight-fitting cap (Figure 1C,D) was constructed for this hole by cutting a 9.5 cm diameter disc out of 2.5 cm thick foam board (Figure 1(D5)) and hot gluing this to an 11 cm disc made of 6 mm thick hardboard (Figure 1(D3)). A 33 cm long strip of insulation foam (Figure 1(D4)) was then hot glued around the circumference of the foam board disc just below where it attached to the hardboard disc. A wall anchor (Figure 1(D2)) was inserted into the center top of the wood disc and a short screw (Figure 1(D1)) was twisted into the anchor to serve as a handle when removing the cap from the container. To operate the lid, it was positioned with the sample access hole at the end furthest opposite the collection bottle (Figure 1G,H). Although the box lid was held in place by a lip seal along its edges, to ensure a tight fit, we stretched a 26 cm long hooked elastic cord (bungee cord) (Figure 1(G5)) from side to side and across the top when samples were in the photocollector (Figure 1G,H).

A portable SMD LED disk light (Figure 1(G3)) (diameter: 7 cm, height: 2.5 cm, color: white, weight: 32 g, lumens: 50, power source: 3 × AAA batteries) was placed into the toe of a dark sock that we fitted tightly over the collection bottle (Figure 1(G4)). The sock and the light were then stretched over the bottle with the light illuminating the collection bottle (Figure 1H).

### 2.2. Operation

To transfer the whole contents of the sweep net, including foliage, debris, and insects, the bottom of the sweep net is converted through the top hole into the box. All insects and debris are pushed through the hole along with the inverted net before carefully withdrawing the sweep net upward through the access hole. The box is sealed by placing the sample access hole cap back into position. The lid should be securely placed atop the collection bin, the collection bottle should be securely threaded onto the container end, and the dark sock should be placed over the collection bottle with the LED light installed but turned off. Once the top cap is secure, the light can be activated, stimulating the insects in the debris to either fly or crawl toward the illuminated collection bottle. More details about the operation, partial construction, and transferring of collected samples from the sweep net to the photocollector can be found at URL (accessed on 24 October 2024): https://youtu.be/9g9jiYxGr-Q.

For most effective operation, the photocollector should remain level with minimal disturbance (although careful transport is possible) and the light should be activated for 1 to 4 h (although see below).

### 2.3. Timed Insect Movement Trials

We tested the efficacy and consistency of the photocollector with insect specimens collected in Grasslands National Park, Saskatchewan, Canada (49.088778, −106.785389) on 28 June 2022. This is a northern mixed grass prairie landscape dominated by native grasses and forbs and only scattered shrubs and trees.

Insects were sampled along 30 transects, each consisting of 80 sweeps (40 steps, approximately 40 m), using a standard sweep net (diameter: 63.5 cm, handle length: 122 cm). The transects were walked at a steady pace to cover the transect distance, and the sweep net was swung in a 180° arc side to side in consecutive steps. The net was angled horizontally and positioned so that its opening was held within, or just above, the vegetation, such that top of the vegetation was within the net or the net was swept just above the ground surface.

Once a transect was complete, the sweep net contents were emptied into the photocollector following the operational protocol described above. For each trial, the collection bottle from each photocollector (containing the sample from one individual transect) was removed after one hour of operation, capped, and labeled. A clean collector bottle, with the light covered by a sock, was immediately installed on the photocollector for subsequent collection during hours 2, 3, and 4 of the 4 h trial.

Once removed from the photocollector, the collection bottle was capped with a 2 cm × 2 cm piece of cloth saturated with Vapona^®^ killing reagent sourced in Ottawa, ON, Canada. All insects appeared to have been killed after 15–30 min, at which time the killing reagent cloth was removed from the cap. After 4 h of operation, all remaining insects and debris in the black container of the photocollector were emptied into a plastic bag which was then placed in a −20 °C freezer for 24 h. All insects were then carefully sorted from the debris. All insects were dried, pinned, counted, labeled, given a unique reference number, and deposited and recorded in the Canadian National Collection of Insects, Arachnids and Nematodes (database accessed at URL (accessed on 15 June 2023): https://cnc.agr.gc.ca/.

We tested whether there were differences among time intervals and taxa using mixed-effects analysis of variance model (RStudio [21]; package “lme4”). Post-hoc Tukey’s tests were used to identify among-group differences. Comparisons were run to test whether there were differences at the order taxonomic level whether and how quickly the insects moved into the illuminated collection bottle as well as whether family differences existed within the orders Diptera and Hymenoptera. Our analyses included only taxa with enough specimens to calculate a reliable mean squares in ANOVA tests among taxa. Three orders had small sample sizes (<5 specimens), which could not produce reliable results, so our minimum sample size was 125, the number of specimens in the next smallest order. This minimum sample size removed Araneae, Neuroptera, Odonata, and Thysanoptera from the order-based analysis, representing 0.11% of specimens across all time intervals, and prevented very rare taxa from excessively influencing the proportional movement rates. Similar criteria were used for families within Diptera (7%) and Hymenoptera (8%), where the threshold was set to 12 specimens (Table 1). RStudio version 2023.12.0 was used for statistical analysis and data visualization (ggplot2 at URL (accessed on 11 June 2023): https://ggplot2.tidyverse.org)) [21]. Summary statistics were calculated using the package “stats”.

## 3. Results

A total of 9830 arthropod specimens belonging to 10 orders were sampled, including Hemiptera 58.2%, Diptera 21.4%, Orthoptera 7.0%, Hymenoptera 4.7%, Coleoptera 3.5%, Araneae (spiders) 3.3%, Lepidoptera 1.7%, Odonata 0.1%, Thysanoptera 0.1%, and Neuroptera 0.1%. From these samples, 73.1% of the total insects moved toward the collection bottle of the photocollector during the first hour, 4.7% in the second hour, 1.4% in the third hour, 0.90% in the fourth hour, and 20.0% did not move into the collection bottle. In total, 1592 specimens were damaged (16.2%) and thus could most likely not move; only 378 (3.9%) specimens that did not move were undamaged. A list of identified families collected in this study is included in Table 1.

Three orders were removed from the analyses as they composed only 0.2% of the overall sample: Thysanoptera, Neuroptera, and Odonata. Because our samples were dominated by Hemiptera, for analyses, we converted real abundances to the proportion of specimens in that family that moved per time interval. Appropriate for the parametric analysis of proportional data, we used an arcsine square root transformation to remove heteroscedasticity and normalize distributions.

For species in all orders, the greatest movement occurred during the first hour of light illumination (mean = 78% of undamaged specimens moved, range = 48% (Lepidoptera) to 92% (Coleoptera); Figure 2). We found significant differences among orders, clustered in three overlapping groups (F_6202_ = 12.71, *p* < 0.0001; Figure 3): Coleoptera, Diptera, Hemiptera, and Orthoptera were all fast movers, Araneae and Hymenoptera were slightly slower, but somewhat indistinguishably, and Lepidoptera were distinctly the slowest (Figure 3).

During the second hour, the composition of fast and slow-moving taxa changed as did the proportion of all taxa remaining in the photocollector. Hemiptera, Hymenoptera, Lepidoptera, and Orthoptera were all relatively fast movers while Araneae, Coleoptera and Diptera were slow (F_6203_ = 5.56, *p* < 0.001; Figure 4). During the third hour, with relatively few specimens remaining that had not moved, the pattern became less distinct. In general, Araneae, Hemiptera, and Hymenoptera were fast movers, while Coleoptera, Diptera, Lepidoptera, and Orthoptera were slow movers (F_6202_ = 4.33, *p* = 0.001; Figure 4). During the fourth hour, no more than 2% of any order moved into the lit collection bottle, but differences among orders remained (F_6202_ = 5.48, *p* < 0.001). Hemiptera and Hymenoptera were fast movers while the other orders were slow. We detected no Lepidoptera moving during this interval (Figure 4).

At the end of the collection period, we sorted through the debris and collected all taxa that had not moved and found large differences among orders (F_6406_ = 9.58, *p* < 0.001; Figure 4). Lepidoptera was the most abundant taxon that did not move toward the lit collection bottle during the four hours of the trial (almost 40% did not move). Araneae, Diptera, and Hymenoptera were the next most abundant orders to not move (between 10 and 15% remaining), while there were very few specimens of Coleoptera, Hemiptera, and Orthoptera that did not move (5% or less).

We identified Hymenoptera and Diptera to the family level to explore within-order patterns for orders with clear phototaxis. Within those orders, not every family was sufficiently abundant to include in our analyses, but we could include six families of Hymenoptera and 12 families of Diptera (Table 1).

Consistent with the patterns we observed for all orders, from 60 to 100% of specimens in families of Hymenoptera moved in the first hour. While some families, like Figitidae and Ichneumonidae, showed strong and immediate phototaxis, others like Encyrtidae and, to some extent, Ichneumonidae, showed a flatter response with greater proportions of specimens taking four hours to move. Although families of Hymenoptera differed in movement during the four hours in the trial (F_5110_ = 28.5, *p* < 0.001), the interaction of family and hour was not significant (*p* = 0.11). This suggests that the variable patterns we observed (Figure 5) still reflected a consistent overall pattern for Hymenoptera. Although on average 65% of Formicidae moved in the first hour, there was a large proportion that did not move toward the light during the 4 h trial.

Diptera responded differently than Hymenoptera. We found strongly significant interactions of hour of movement and Diptera (F_7207_ = 5.5, *p* < 0.0001). Post-hoc tests indicated that this was largely due to the temporal patterns of Agromyzidae, Sepsidae, and Chloropidae, which showed a surge of movement after two hours (Figure 6). An analysis of the proportion of families of Diptera that did not move toward the light after four hours showed significant variation (F_1185_ = 6.41, *p* < 0.0001). Unlike with Hymenoptera, the patten for Diptera was largely driven by Anthomyiidae, which tended to move far earlier than other families in the order.

## 4. Discussion

Our findings confirmed what others have assumed—the photocollector device efficiently extracted insects from field debris between 1 and 4 h of operation. Particularly in the context of biodiversity studies involving large insect populations, this can provide significant advantages. The photocollector eliminates the need for labor-intensive manual collection of insects from sweep net specimens, thereby saving both time and costs in the research process. The device’s lightweight design and cost-effectiveness render it a valuable and practical instrument for taxonomists and ecologists to use in the field.

Our experiments demonstrate that the photocollector collected all examined orders and families, including spiders (Araneae). Although spiders have negative phototropism, they moved toward the collection bottle in the photocollector. They were, however, the second most abundant order found in the debris that had not moved after 4 h. Most of the spider families we observed were web-builders that build webs around a light source at night to catch prey, suggesting that they were motivated by predation rather than behavioral phototaxis.

Lepidoptera larvae were by far the most reluctant movers among our grassland specimens. Most of the families of Lepidoptera collected by sweep net in this study belonged to Scythrididae. These are mostly diurnal animals and are not strongly drawn to light sources [22,23]. Sweep netting is not an appropriate method for collecting Lepidoptera due to wing damage [9]; leaving them in a photocollector leads to additional damage from other insects.

The photocollector had the potential to gather most insects and spiders within 1–2 h. Although insects continued to move toward the collection bottle after 2 h, longer durations can affect sample quality. The risk of specimen damage from larger, heavier, or hard-shelled insects (e.g., grasshoppers, Coleoptera, Orthoptera) increases with the time they spend active in the collection bottle. Despite 95% of specimens transferring to the main bottle after two hours, opting to collect 90% of specimens within one hour may be preferable to increased likelihood of specimen damage. Further research with short intervals within the first hour is warranted.

Families within Diptera demonstrated more consistent phototaxis than families within Hymenoptera, with consequences for using a photocollector in studies focused on these taxa, which is supported by the findings of Sheba et al. [24] who noted that flies (Diptera) exhibit strong phototropism in comparison with all other orders. While Figitidae consistently moved toward the light in the first hour, families of Hymenoptera, like Encyrtidae and Pteromalidae, did not, showing a weaker tendency for rapid movement toward the light. Among the families of Diptera, this inconsistency was only strongly evident in Agromyzidae and weakly in Sepsidae and Chloropidae. Hence, care should be taken in studies to confirm specimen movement rates during biodiversity studies.

Although our insect collections were made entirely within the mixed grass biome, we feel that our trials, and hence the photocollector, has broader applicability. Most insects across biomes demonstrate some sort of phototaxis, and physical collection methods, like sweep netting, continue to be one of the most effective ways to collect a reliable cross-section of the insect biota. Hence, a tool that uses light to sort out debris from live insect specimens has broad utility. We have demonstrated that a simple, portable, and inexpensive photocollector can quickly and efficiently sort live insect specimens from plant debris during field collections.

## Figures and Tables

**Figure 1 insects-15-00896-f001:**
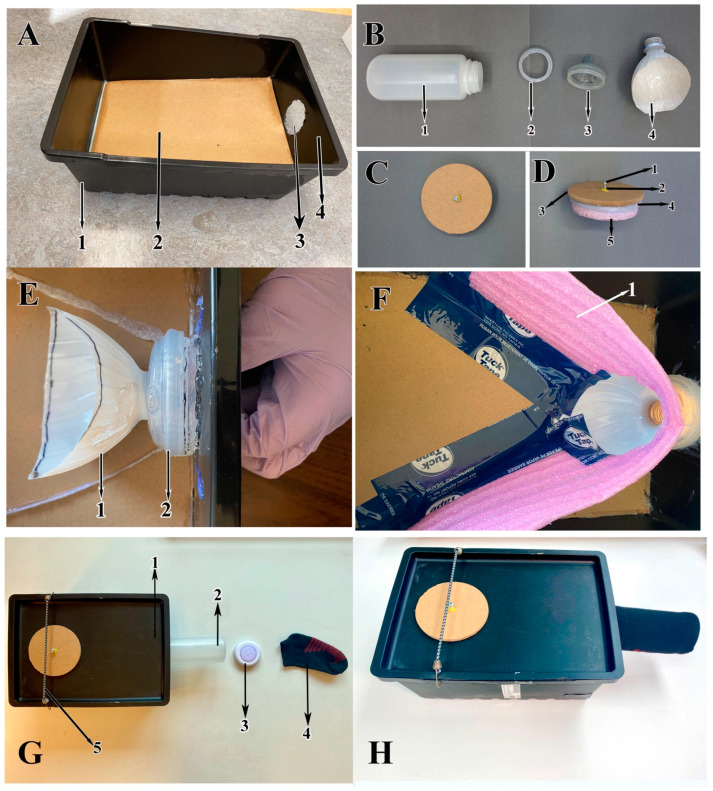
Photocollector: (**A1**) black plastic box (Ikea Uppsnofsasd™), (**A2**) wooden ramp, (**A3**) collector bottle entrance hole, (**A4**) front wall, (**B1**) collection bottle, (**B2**) threaded bottle cap (top of the cap has been cut off), (**B3**) calf feeder nipple, (**B4**) water bottle head (cut and painted). (**C**,**D**) Photocollector box cap (**C**) (dorsal view) and (**D**) (lateral view): (**D1**) wood screw, (**D2**) wall anchor, (**D3**) wooden disk, (**D4**) insulation foam stripe, (**D5**) foam board. (**E**) Gluing the threaded cap covered with calf feeder nipple to the collector bottle entrance hole: (**E1**) water bottle head (cut and painted), (**E2**) calf feeder nipple covering the threaded cap. (**F**) Interior with all components: (**F1**) guide wall foam. (**G**,**H**) Exterior: (**G1**) plastic box lid, (**G2**) collection bottle, (**G3**) LED light, (**G4**) sock; (**G5**) bungee cord, (**H**) assembled photocollector.

**Figure 2 insects-15-00896-f002:**
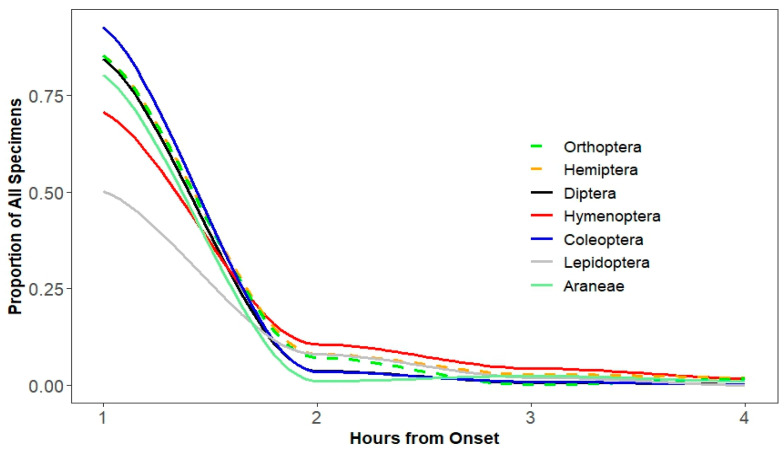
Temporal distribution by order of the proportion of arthropod specimens moving into a lit collection bottle from a dark photocollector. Collections were made at hourly intervals for 4 h. Proportions include the specimens that did not move toward the light.

**Figure 3 insects-15-00896-f003:**
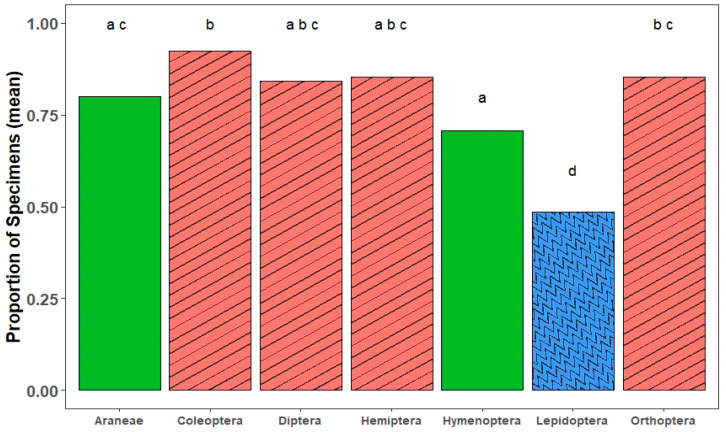
Mean proportion of specimens that moved during the first hour of photocollector operation for each order. Fill patterns group insect orders into three general movement rates (red striped = fast, green solid = moderate, blue zigzag = slow). Bars with different lowercase letters are orders with statistically different proportions of individuals moving in the first hour. Analysis based on ANOVA with a post-hoc Tukey’s test (*p* < 0.05).

**Figure 4 insects-15-00896-f004:**
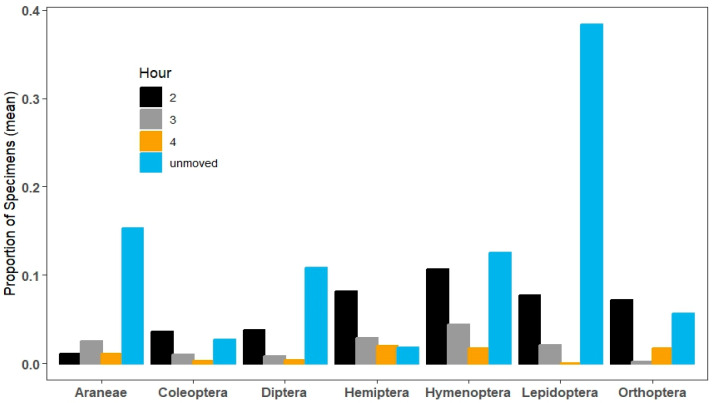
Proportion of specimens moving after the first hour by insect order. Black bars indicate movements in the second hour, gray bars indicate movements in the third hour, orange bars indicate movements in the fourth hour, and blue bars indicate the proportion of specimens that did not move after four hours. For most orders (Hemiptera was an exception), specimens that did not move in the first hour tended to not move at all (blue bars), but there were significant patterns among taxa in movement rates in hours 2, 3, and 4. Proportions of Hemiptera and Hymenoptera moving in hour 2 were far greater than those of Araneae and Coleoptera (Tukey’s post-hoc test: *p*< 0.05), while a greater proportion of Orthoptera moved in hour 2 than Araneae (Tukey’s post-hoc test: *p* = 0.02). Orthoptera moved significantly less than Hemiptera (Tukey’s post-hoc test: *p* < 0.002) and Hymenoptera (Tukey’s post-hoc test: *p* < 0.02) in hour 3. During hour 4, more Hemiptera moved than Araneae, Coleoptera, Diptera, Lepidoptera, and Orthoptera (Tukey’s post-hoc test: *p* < 0.02). Unmoved specimens were dominantly Lepidoptera (*p* < 0.05), while a lower proportion of Coleoptera did not move compared to all orders, except for Araneae and Orthoptera (*p* = 0.01).

**Figure 5 insects-15-00896-f005:**
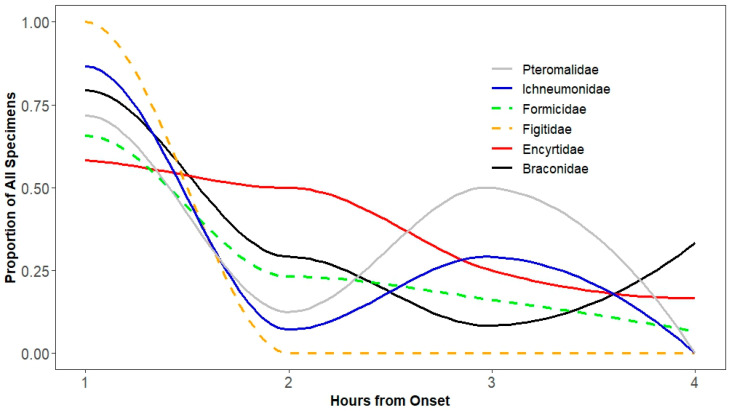
Temporal distribution of families in the order Hymenoptera moving into a lit collection bottle from a dark photocollector. Collections were made at hourly intervals for 4 h. Proportions include the specimens that did not move toward the light after 4 h.

**Figure 6 insects-15-00896-f006:**
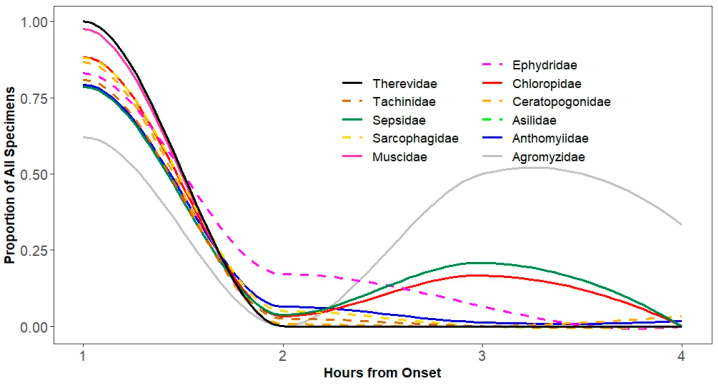
Temporal distribution of Families in the Order Diptera moving into a lit collection bottle from a dark photocollector. Collections were made at hourly intervals for 4 h. Proportions include the specimens that did not move to the light after 4 h.

**Table 1 insects-15-00896-t001:** Taxa collected during sweep net trials. Families within Hymenoptera and Diptera were further identified for family-specific movement analysis. We indicate whether orders (≥125) or families (≥12) had sufficient specimens for analyses.

Order	Family	Analyzed
Hemiptera	n/a ^1^	yes
Orthoptera	n/a	yes
Coleoptera	n/a	yes
Lepidoptera	n/a	yes
Odonata	n/a	no
Araneae	n/a	no
Thysanoptera	n/a	no
Neuroptera	n/a	no
Hymenoptera		yes
Diptera		yes
Hymenoptera	Andrenidae Latreille, 1802	no
Hymenoptera	Apidae Latreille, 1802	no
Hymenoptera	Braconidae Nees, 1811	yes
Hymenoptera	Chalcidoidea Latreille, 1817	no
Hymenoptera	Crabronidae Latreille, 1802	no
Hymenoptera	Diapriidae Haliday, 1833	no
Hymenoptera	Encyrtidae Walker, 1837	yes
Hymenoptera	Eulophidae Westwood, 1829	no
Hymenoptera	Eurytomidae Walker 1832	no
Hymenoptera	Figitidae Thomson, 1862	yes
Hymenoptera	Formicidae Latreille, 1802	yes
Hymenoptera	Ichneumonidae Latreille 1802	yes
Hymenoptera	Megachilidae, Latreille, 1802	no
Hymenoptera	Pteromalidae Dalman 1820	yes
Hymenoptera	Scelionidae Haliday, 1839	no
Hymenoptera	Torymidae Walker 1833	no
Hymenoptera	Xiphydriidae Leach, 1819	no
Diptera	Agromyzidae Fallen, 1823	yes
Diptera	Anthomyiidae Robineau-Desvoidy, 1830	yes
Diptera	Asilidae Latreille, 1802	yes
Diptera	Bombyliidae Latreille, 1802	no
Diptera	Cecidomyiidae Newman, 1835	no
Diptera	Ceratopogonidae Newman, 1834	yes
Diptera	Chamaemyiidae Hendel, 1910	no
Diptera	Chironomidae Newman, 1834	no
Diptera	Chloropidae Rondani, 1856	yes
Diptera	Conopidae Latreille, 1802	no
Diptera	Culicidae Meigen, 1818	no
Diptera	Dolichopodidae Latreille, 1809	no
Diptera	Empididae Latreille, 1804	no
Diptera	Ephydridae Zetterstedt, 1837	yes
Diptera	Heleomyzidae Westwood, 1840	no
Diptera	Hybotidae Macquart, 1823	no
Diptera	Muscidae Latreille, 1802	yes
Diptera	Pipunculidae Walker, 1834	no
Diptera	Sarcophagidae Macquart, 1834	yes
Diptera	Scathophagidae Robineau-Desvoidy, 1830	yes
Diptera	Sepsidae Walker, 1833	yes
Diptera	Simuliidae Olfers, 1816	no
Diptera	Syrphidae Latreille, 1802	no
Diptera	Tachinidae Robineau-Desvoidy, 1830	yes
Diptera	Tephritidae Newman, 1834	no
Diptera	Therevidae Newman, 1834	yes
Diptera	Ulidiidae Macquart, 1835	no

^1^ not applicable: These orders not analyzed at the family level.

## Data Availability

The data presented in this study are available upon request from the corresponding author.

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
