# Peer review of "A Portable Photocollector for the Field Collection of Insects in Biodiversity Assessment"

_insects, 2024, doi:10.3390/insects15110896_

Round 1
Reviewer 1 Report
Comments and Suggestions for Authors
I have finished with the evaluation of the manuscript titled “Testing a portable photocollector for the field collection of insects in biodiversity assessment”
This manuscript is the result of focused studies on the development and introduction of a photocollector device by researchers who have recently completed an extensive sampling program in the area and have collected and prepared a vast amount of samples for biodiversity studies. Given the importance of this sampling method across a wide range of biodiversity studies, it is expected that the tools introduced in this work will be widely utilized by other researchers or that more optimized models will be developed based on it. Ultimately, if published, the article will receive numerous citations. So, I recommend publishing this work after a moderate revision (mainly on the writing style).
The evaluation results are as follows, and the authors should address these issues before publication. [The main points have been noted as comments in the original text]
1- The title of the manuscript is acceptable with minor revisions (as noted).
2- In the abstract section, some content has been presented inconsistently or ambiguously and needs to be corrected. This issue is also observed in other sections of the article to varying degrees.
3- The scope and significance of insect sampling using the netting method should be thoroughly explained by reviewing a larger number of references. The same applies to references regarding negative phototropism (Berelse–Tullgren funnel).
4- The phenomenon of positive phototropism (the basis for the development of the new device) is a broad topic, and only one aspect of it is connected to creating a light gateway for the insects that are inherently active during the day, This topic should be clarified as much as possible in the introduction section of the manuscript.
5- In some parts of the Materials and Methods section, the authors have recommended the steps of the work as a protocol to the reader. However, this section should be presented in a passive (past participle), and the recommendations should be moved to the end of the "Discussion" section.
6- Other points are scattered but numerous throughout the text, and the authors should carefully and consistently address them. The recommendations provided will help enhance the quality of the work and also improve the credibility of the article's practical sections. The authors are invited to incorporate them into the final version of their article carefully.
I hope this valuable article will be published soon and made available to interested readers.

Some parts were technically or linguistically unclear to me (as a reviewer). So, I tried to mark these sections on the PDF file. The authors are recommended to re-write these sections and clarify the statements.
Author Response
Thank you very much for taking the time to review this manuscript. Please find the detailed responses below and the corresponding revisions/corrections highlighted/in track changes in the re-submitted files.
reviewer 1:
Line 41:
Comment: what about the rest?
Answer: We only tried to motion the main result here and explained details in the result section since based on the journal policy, the abstract should be a total of about 200 words maximum. Therefore, we had to remove some words that you mentioned in line 32 and shorten the abstract.
Line 41:
Comment: various group of?
Answer: We mentioned “in when” and then “families of the Diptera and Hymenoptera moved”. As we considered different times among families of Diptera and Hymenoptera, we believe the current sentence can explain this matter.
Line 103:
Comment: where is the light source?
Answer: G3
line 202:
Comment: unclear to me.
Answer: It says the photocollector was kept level, mostly horizontally, to allow insects to move toward the light.
Line 215:
Comment: Any reference for some sections of this method? or all is original?
Answer: It is original and designed based on the environmental conditions in grassland.
Line 344:
Comment: What is the significance of the speed of movement? In my opinion, it is important for the separation of different groups, …
Answer: The movement comparison helped assess the efficiency of the photocollector and identify the optimal time for different insect groups, which is fully mentioned in the last paragraph of the introduction and the result section.
The rest of the modifications are marked in red font in the manuscript.
-------
Other comments:
Comment 1- The title of the manuscript is acceptable with minor revisions (as noted).
-Answer: It is done.
Comment 2- In the abstract section, some content has been presented inconsistently or ambiguously and needs to be corrected. This issue is also observed in other sections of the article to varying degrees.
-Answer: It is corrected.
Comment 3- The scope and significance of insect sampling using the netting method should be thoroughly explained by reviewing a larger number of references. The same applies to references regarding negative phototropism (Berelse–Tullgren funnel).
-Answer: The relevant references are added regarding using of the sweep net in different habitats as well as negative phototropism reference. We believe that the main goal of the manuscript should be to introduce the photocollector and its efficiency.
Comment 4- The phenomenon of positive phototropism (the basis for the development of the new device) is a broad topic, and only one aspect of it is connected to creating a light gateway for the insects that are inherently active during the day, this topic should be clarified as much as possible in the introduction section of the manuscript.
-Answer: It is added.
Comment 5- In some parts of the Materials and Methods section, the authors have recommended the steps of the work as a protocol to the reader. However, this section should be presented in a passive (past participle), and the recommendations should be moved to the end of the "Discussion" section.
-Answer: It is corrected.
Comment 6- Other points are scattered but numerous throughout the text, and the authors should carefully and consistently address them. The recommendations provided will help enhance the quality of the work and also improve the credibility of the article's practical sections. The authors are invited to incorporate them into the final version of their article carefully.
-Answer: The rest of the modifications are marked in red font in the manuscript.
Reviewer 2 Report
Comments and Suggestions for Authors
This is a well written manuscript outlining an interesting insect collecting device. My comments are written directly on the pdf version in comment boxes.

Author Response
Thank you very much for taking the time to review this manuscript. Please find the detailed responses below and the corresponding revisions/corrections highlighted/in track changes in the re-submitted files.
Reviewer 2:
Line 83:
Comment: Should check the proceedings volume available in CNC - Diptera.
Answer: We looked for the proceedings in CNC and couldn’t find it.
Line 205:
Comment: ensure this subheading is linked up with the text and not on a separate page (in this journal, it is up to the authors for accurate formatting. Editors are not very helpful.
Answer: It is linked to the text.
Line 425:
Comment: Are larvae positive phototropic?
Answer: No, they are not.
The rest of the modifications are marked in red font in the manuscript.
Reviewer 3 Report
Comments and Suggestions for Authors
The aim of the publication is very straightforward and succeeds in showing that the photocollector is an efficient tool to be used in the field to sort out the invertebrate samples from other debris collected through net sweeping. The online video really helped clarify some of the assembly steps and I would recommend adding time tags on the video so the reader can refer to the exact points of assembly of the photocollector.
Just a few comments:
1. The sentence between lines 236 and 238 could be rewritten to make it clearer what was the reasoning to reach these cut-off numbers to have the taxa included in the analysis.
2. In 'Author Contributions' please correct C.K. to C.C.; not sure who J.F. is in the list
Author Response
Thank you very much for taking the time to review this manuscript. Please find the detailed responses below and the corresponding revisions/corrections highlighted/in track changes in the re-submitted files.
Reviewer 3:
Comments:
- The sentence between lines 236 and 238 could be rewritten to make it clearer what was the reasoning to reach these cut-off numbers to have the taxa included in the analysis.
-Answer: change to the following text: Our analyses included only taxa with enough specimens to calculate a reliable Mean Squares in ANOVA tests among taxa. Three orders had small sample sizes (<5 specimens), which could not produce reliable results, so our minimum sample size was 125, the number of specimens in the next smallest order. This minimum sample size removed Araneae, Neuroptera, Odonata, and Thysanoptera from the Order-based analysis, representing 0.11% of specimens across all time intervals and prevented very rare taxa from excessively influencing proportional movement rates. The same criteria were used for Families within Diptera (7%) and Hymenoptera (8%) where the threshold was set to 12 specimens (Table 1).
- In 'Author Contributions' please correct C.K. to C.C.; not sure who J.F. is in the list
-Answer: It is corrected.